# 10-Year Rotavirus Infection Surveillance: Epidemiological Trends in the Pediatric Population of Perugia Province

**DOI:** 10.3390/ijerph17031008

**Published:** 2020-02-05

**Authors:** Chiara de Waure, Laura Sarnari, Manuela Chiavarini, Giovanni Ianiro, Marina Monini, Anna Alunno, Barbara Camilloni

**Affiliations:** 1Department of Experimental Medicine, University of Perugia, 06132 Perugia, Italy; chiara.dewaure@unipg.it (C.d.W.); sarnari13@gmail.com (L.S.); 2Department of Food Safety, Nutrition and Veterinary Public Health, Istituto Superiore di Sanità, 00161 Rome, Italy; giovanni.ianiro@iss.it (G.I.); marina.monini@iss.it (M.M.); 3Department of Medicine, University of Perugia, 06132 Perugia, Italy; anna.alunno@unipg.it (A.A.); barbara.camilloni@unipg.it (B.C.)

**Keywords:** rotavirus infection, pediatric population, epidemiology

## Abstract

Rotavirus (RV) infections are a leading cause of severe gastroenteritis in children, and vaccination is currently recommended in Italy, according to the National Immunization Plan 2017–2019. The objective of this study was to describe the epidemiological and molecular RV surveillance in the pediatric population of Perugia province, Umbria. Between September 2007 and August 2018, 663 RV-positive stool specimens were collected from children <15 years of age presenting with gastroenteritis to the emergency room of the Perugia province hospitals who were then hospitalized. Yearly hospitalization rates were expressed per 100,000 persons, and denominators were extrapolated from the National Institute of Statistics. During the 10-year surveillance, the epidemiological trend was fluctuating but slightly decreasing (Max: 89.7 per 100,000 in 2010/2011; Min: 34.8 per 100,000 in 2017/2018). The hospitalization rate was higher in males and in children under five years of age. Among common genotypes, G1P[8] was prevalent most of the years. The uncommon G12P[8] genotype emerged and was the most common in 2012/2013 (58.2%). Afterwards, its circulation remained high. As the Umbria Region started vaccinating from the 2018 birth cohort, our study reviewed pre-vaccination data and will help to assess the protection induced by vaccination and its effect on circulating strains.

## 1. Introduction

Rotavirus (RV) infections are a leading cause of severe acute gastroenteritis (AGE) in children [1]. Worldwide, in 2016, RV accounted for most episodes of diarrhea among children under the age of five, representing the leading cause of mortality from diarrhea (128,515 deaths, 20.3 deaths per 100,000 in children under the age of five) [2]. In the European Union, RV annually accounts for 3.6 million episodes of gastroenteritis, 700,000 outpatient visits, 87,000 hospitalizations, and 231 deaths among children younger than five years of age [3].

RV is an 11-segment double-stranded RNA virus encoding six structural proteins (VP1–VP4, VP6, VP7) and five or six non-structural proteins (NSP1–NSP5/6) [4]. Based on antigenic differences of the VP6 protein, RVs are classified into groups from A to J, whereas they are classified into G-genotypes (VP7, glycoprotein) and P-genotypes (VP4, protease-sensitive), respectively [5,6], based on the sequence of VP7 and VP4 genes. Worldwide, the majority of RVs causing diarrhea in children belong to serogroup A (RVA) [7]. The most common G/P genotype combinations causing 90% of infections in humans are G1P[8], G2P[4], G3P[8], G4P[8], G9P[8], and G12P[8] [8,9,10,11]. However, several unusual, rare, or novel strains have been generated from time to time by mechanisms that may include reassortment due to coinfection with human and animal strains [12,13,14]. Changes in circulating RVA strains can also occur through natural selection [15] or because of vaccine selection pressure [16]. In fact, in the last decade, RVA vaccination started to be promoted because of the health and economic burden linked to RVA infection [17].

In 2009, the World Health Organization recommended the inclusion of RVA vaccines (Rotarix, monovalent, containing a live-attenuated human strain; and RotaTeq, pentavalent, containing five human–bovine reassortant live attenuated strains) in the Extended Program on Immunization [18,19].

In Italy, Sicily started offering the RVA vaccination actively and free of charge from May 2012. Other Italian Regions (Apulia, Friuli Venezia Giulia, Veneto, and Piedmont) began offering the RVA vaccination free of charge to children at risk from 2014 [20]. Eventually, RVA vaccination for all newborns was recommended nationwide by the Italian National Immunization Plan 2017–2019. Following this, the Umbria Region introduced the universal RVA vaccination starting with the 2018 birth cohort. Nevertheless, continuous epidemiological surveillance is needed to identify the protection engendered by the vaccination and the effect that it may have on circulating strains.

To the best of our knowledge, another Italian study reported the results of a 27-year-long surveillance of RV infections [21]. The study summarized local surveillance data and provided an in-depth investigation of the temporal pattern of RVAs variation in the pre-vaccine era contributing to our understanding of whether the RVA vaccine could alter the epidemiology of RVAs. Similarly, a recent study described RVA genotypes circulating in Italy in the three years prior to the introduction of vaccination in the Italian National Immunization Plan [22].

The objective of this study is to report epidemiological and molecular RVA surveillance data collected in the pediatric population of the Perugia province in the Umbria Region of Italy. As the study includes pre-vaccination data, it will provide relevant information for researchers and public health authorities for comparative analysis with post-vaccination data, and will eventually help to assess the efficacy of vaccination and its potential selective pressure.

## 2. Materials and Methods

This was a population-based surveillance study performed in the Umbria Region from 1 September 2007 to 31 August 2018 that encompassed 11 RVA surveillance seasons as defined by the European Rotavirus Network (EuroRotaNet). The Umbria Region, which has a resident population of 882,015 inhabitants (as of 1 January 2019) [23], has participated as a reference center within the national surveillance of RotaNet-Italy since 2007 [24].

RotaNet-Italy is part of EuroRotaNet, a network of European laboratories that involves 16 countries and is aimed at studying the molecular epidemiology of RVA.

The EuroRotaNet protocol requires the collection of a stool sample from patients hospitalized for acute gastroenteritis (AGE) and that it be examined locally using routine commercial diagnostic tests. After the first diagnosis through commercial antigen detection methods (enzyme immunoassay—EIA), RVA-positive samples are frozen (<−20 °C), stored, and sent to the regional reference center of the University of Perugia. The frozen samples are then sent to the Istituto Superiore di Sanità (ISS), where the molecular characterization of RVA-positive samples by RT-nested-PCR tests for genes 4 (VP4) and 9 (VP7) is carried out according to standardized European protocols (https://www.eurorotanet.com/) in order to identify the binary G/P genotype. For each case, clinical and epidemiological information is collected through a questionnaire filled in by the pediatrician or doctor, in accordance with the rules on informed consent at the collaborating facilities [24,25]. For this study, only data relating to the pediatric population were evaluated.

Children below 15 years of age suffering from AGE, who accessed the emergency room of the hospitals of Perugia province (namely, seven hospitals in the towns of Assisi, Castiglione del Lago, Città di Castello, Foligno, Gubbio, Pantalla, and Perugia) were considered eligible for inclusion in the study if they were then admitted at the hospital as inpatients, including one-day care admissions (at least one overnight stay). Nosocomial infections, namely, those occurring in inpatients at least 48 hours after hospital admission, were excluded, as were cases requiring emergency medical consultations not followed by inpatient hospitalization.

To evaluate the epidemiological burden of RVA infections in Perugia, the hospitalization rate for 100,000 inhabitants was calculated for each year. RVA-positive stool specimens were considered as the numerator, whereas the pediatric population below 15 years of age resident in the province of Perugia was the denominator. This last figure was calculated as the average between the population estimate for 1 January and 31 December of each year. Data on pediatric resident population were collected from the Demo Istat [23], stratified by age class and sex. Hospitalization rates were calculated overall and with respect to sex and the age group from 0 to 5 years of age, which is the most affected by AGE due to RV.

Data were reported by means of descriptive statistics.

## 3. Results

During the study period, 663 RVA-positive stool specimens were collected and genotyped for VP7 and VP4.

Of the whole sample, 59.1% (392/663) were collected from male children and the mean age of RVA-positive patients was 29.2 months (±30.2); 88.8% (589/663) of the samples were collected from children under five years of age.

The hospitalization rate reached a maximum of 89.7 for 100,000 inhabitants in the 2010/2011 season and a minimum of 34.8 per 100,000 inhabitants in 2017/2018 (Table 1). The hospitalization rate was higher in males (Max: 113.9 per 100,000 in 2010/2011 and Min: 44.5 per 100,000 in 2017/2018) compared to females (Max: 70 per 100,000 in 2014/2015 and Min: 24.6 per 100,000 in 2017/2018) (Table 1).

In children aged five years or less, the hospitalization rate was higher as compared to the whole sample (Max: 222.1 cases per 100,000 in the 2010/2011 season and Min: 81.6 per 100,000 in the 2017/2018 season), with males showing higher figures than females (Max: 289.2 in 2010/2011 and Min: 115.2 in 2015/2016 in male children vs. Max: 180.1 in 2011/2012 and Min: 44.9 in 2017/2018 in female children; Table 1). Nevertheless, the hospitalization rate was higher in females than in males in the 2011/2012 season (180.1 vs. 170.2 per 100,000).

With respect to genotyping, non-typeable RVA strains and mixed infections were identified in 2.9% (19/663) and in 6.2% (41/663) of cases, respectively. The G1P[8] genotype was identified in 268 out of 663 samples (40.4%), being the prevalent genotype in most of the surveillance seasons (7 out of 11; 63.6%). In particular, it reached a maximum of 70.2% (59/84) of all genotyped RVA in 2010/2011, whereas it was not detected at all in the 2016/2017 season. Occasionally, other genotypes were found in greater proportion. In particular, G9P[8] was found in 92 out of 663 samples (13.9%) with a maximum of 44.7% (17/38) in the 2015/2016 season, whereas G4P[8] was detected in 89 out of 663 samples (13.4% of specimens) with a peak of 44.6% (29/65) in 2011/2012 (Figure 1; Table 2).

For the first time, in the 2012/2013 season, G12P[8] (considered an emerging genotype) was preponderant (58.2%, 32/55), remaining high but discontinuous and fluctuating in the following years as well. From the 2012/2013 season onwards, G12P[8] was found in 28.8% (91/316), while it was previously close to 0% (Figure 1; Table 2). Uncommon genotypes were found in 13 out of 663 samples (2.0%) including G10P[8] (*n* = 4, 0.6%), G2P[8] (*n* = 3, 0.5%), G4P[4] (*n* = 2, 0.3%), G1P[4] (*n* = 1, 0.2%), G4P[10] (*n* = 1, 0.2%), G6P[9] (*n* = 1, 0.2%), and G9P[4] (*n* = 1, 0.2%).

As for mixed infections, 12 different mixed genotypes were identified in 41 samples. Among them, the most common were G1G9P[8] (*n* = 12, 1.8%), G1G2P[4]P[8] (*n* = 11, 1.7%), G1G4P[8] (*n* = 6, 0.9%), G9G12P[8] (*n* = 3, 0.5%), and G2G9P[4]P[8] (*n* = 2, 0.3%). G1G12P[8], G1G2P[4], G1G3P[4]P[8], G2G12P[4]P[8], G2G4P[4], G3G9P[8], and G4G9P[8] were identified in one sample each (0.2%).

## 4. Discussion

This paper analyzed the epidemiological and molecular pattern of gastroenteritis due to RVA (RVAGE) that led to hospitalization in the pediatric population of Perugia province in the Umbria Region. Children under five years of age were the most affected by the disease, and their hospitalization rates were aligned with European data [26]. However, they were slightly lower with respect to available predicted estimates for European countries [3,27] and other Italian data that used hospital discharge forms and RV-specific International Classification of Diseases (ICD) codes in primary and secondary diagnosis [28,29,30,31,32,33], or laboratory confirmation without excluding nosocomial cases [34]. The use of hospital discharge forms may result in misclassification and does not allow distinguishing between community-acquired and nosocomial infections. In fact, Marchetti et al. [31] reported that when considering only RVAGE in primary diagnosis, hospitalization rates fell from 279/100,000 to 158/100,000, in line with our data. Similarly, in another nationwide study the mean hospitalization rate for RVAGE in primary diagnosis was 146/100,000 in children under five years of age [28]. It may be supposed that RVAGE reported in primary diagnosis better reflects the burden of community-acquired RVAGE, but it should be taken into account that a diagnosis of RV is only assigned on the basis of clinical features without any laboratory testing in a well-documented percentage of cases [35]. Our study based on the laboratory confirmation of RV infection gives a useful contribution to the knowledge of the epidemiological features of patients affected by RVAGE in the light of the current spreading of RV vaccination.

In terms of children’s characteristics, hospitalization rates were higher in males than in females, in line with other available evidence [29,30,36]. Additionally, the mean age of cases was close to that of patients with community-acquired RVAGE in the study by Panatto et al. [34] and in line with international data, proving that the incidence peak is observed in the first and second year in Europe [37].

Even though we did not apply any methods to assess the significance of temporal trends, a decreasing trend in hospitalization rates can be appreciated. This could be due to the increasing attention paid to the appropriateness of the management of AGE. In fact, in Italy, the hospitalization rate for AGE in childhood has been used and is currently used as an indicator to assess the quality of primary care.

As for the genotype, the percentage of non-typable samples and mixed infections followed available data [10,22,38] as well as the two most frequently isolated genotypes, namely G1P[8] and G9P[8] [38]. In particular, G1P[8] was also the most prevalent strain during the 27-year-long surveillance performed in Sicily [21]. Nonetheless, G12P[8] was also frequently detected in our study, but recall that this genotype became prevalent in Italy between 2014 and 2017 [22], after its first detection during an outbreak in the Umbria Region in 2012 [39].

Collected data represent a valid working basis for the evaluation of the impact of currently ongoing implementation of RV vaccination in the Umbria Region. In fact, even though this paper does not release a thorough overview of all community-acquired cases, it is expected that vaccination will also affect hospitalizations due to RVAGE. Furthermore, since the paper also looked at age groups other than the one targeted by the vaccination program, it allows future direct and indirect impacts to be evaluated.

This study has some limitations. One is represented by the possibility of either an over- or an underestimation of the hospitalization rates because of potential patient’s referral to other hospitals of the Umbria or nearest regions. Nevertheless, considering that we paid attention to Perugia, where the main hospital of the Umbria Region is located, underestimation could be ruled out, whereas overestimation could still be possible. Another limitation of the study is linked to data used for the analysis that do not allow comprehensive description of the overall burden of RVAGE. In fact, RVAGE can be managed at home or at a primary-care level without any access to the hospital, in particular if affecting people not at risk. Indeed, hospitalization rates do not allow depiction of the overall incidence of the disease but are important to understand the health services utilization determined by RVAGE. Regardless, it should be observed that cases managed outside the hospital do not commonly undergo microbiological ascertainment and could be difficult to assess within a surveillance system. Nevertheless, it should be acknowledged that several indicators are suggested by the European Centre for Disease Prevention and Control to study the impact of RV vaccination and that a confirmed case of AGE due to RV, namely, a child with a positive RV laboratory test, is just one of them [40].

Among the strengths of this study, we can mention the fact that it relies on data collected according to a standardized protocol. Furthermore, it provides a specific point of view on community-acquired RVAGE that led to a hospitalization underlying responsible genotypes during a long period of time (i.e., 10 years). These data represent useful baseline information for evaluating the epidemiological and molecular changes that may occur with the spread of vaccination.

## 5. Conclusions

Our study reviewed the epidemiology of community-acquired RVAGE that led to hospitalization in children, making it the largest continuously conducted RV surveillance in Central Italy. The trend recorded in the pre-vaccination era in the Umbria region was fluctuating with only a slight decrease from the start of the surveillance. The most affected children were those under five years of age and males. During the 10-year surveillance, the emergence of the uncommon genotype G12P[8] was appreciated. These data, although not exhaustively depicting the epidemiology of RVAGE, will be useful for assessing the effectiveness of vaccination in the Umbria Region.

## Figures and Tables

**Figure 1 ijerph-17-01008-f001:**
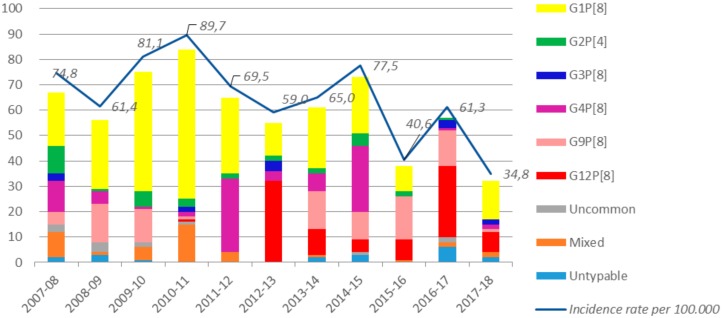
Absolute frequencies of RV genotypes detected (see legend) and hospitalization rate (including one-day care) for RVAGE per 100,000 inhabitants stratified by surveillance season.

**Table 1 ijerph-17-01008-t001:** Hospitalization rate (including one-day care) for RVAGE per 100,000 inhabitants in the 0 to 15-year-old age range.

RV Season	0–15 y	Males 0–15 y	Females 0–15 y	0–5 y	Males 0–5 y	Females 0–5 y
2007–2008	74.8	84.4	64.6	198.9	218.8	177.9
2008–2009	61.4	63.8	58.9	140.1	141.5	138.5
2009–2010	81.1	96.5	64.8	191.9	218.2	164.0
2010–2011	89.7	113.9	64.0	222.1	289.2	150.9
2011–2012	69.5	70.5	68.3	175.0	170.2	180.1
2012–2013	59.0	71.0	46.4	139.1	168.6	108.2
2013–2014	65.0	74.5	54.8	148.3	169.9	125.8
2014–2015	77.5	84.6	70.0	185.4	195.4	175.0
2015–2016	40.6	49.9	30.7	91.3	115.2	66.2
2016–2017	61.3	67.1	55.3	142.6	142.6	142.5
2017–2018	34.8	44.5	24.6	81.6	116.8	44.9

RVAGE: gastroenteritis due to serogroup A rotavirus (RVA).

**Table 2 ijerph-17-01008-t002:** Absolute and relative frequencies of detected RV genotypes stratified by surveillance season.

RV Season	G12P[8]	G1P[8]	G2P[4]	G3P[8]	G4P[8]	G9P[8]	Uncommon	Mixed	Non-Typable	Tot **
2007–08	0 (0.0%)	21 (31.3%)	11 (16.4%)	3 (4.5%)	12 (17.9%)	5 (7.5%)	3 (4.5%)	10 (14.9%)	2 (3.0%)	67 (10.1%)
2008–09	0 (0.0%)	27 (48.2%)	1 (1.8%)	0 (0.0%)	5 (8.9%)	15 (26.8%)	4 (7.1%)	1 (1.8%)	3 (5.4%)	56 (8.4%)
2009–10	0 (0.0%)	47 (62.7%)	6 (8.0%)	0 (0.0%)	1 (1.3%)	13 (17.3%)	2 (2.7%)	5 (6.7%)	1 (1.3%)	75 (11.3%)
2010–11	1 (1.2%)	59 (70.2%)	3 (3.6%)	2 (2.4%)	2 (2.4%)	1 (1.2%)	1 (1.2%)	15 (17.9%)	0 (0.0%)	84 (12.7%)
2011–12	0 (0.0%)	30 (46.2%)	2 (3.1%)	0 (0.0%)	29 (44.6%)	0 (0.0%)	0 (0.0%)	4 (6.2%)	0 (0.0%)	65 (9.8%)
2012–13	32 (58.2%)	13 (23.6%)	2 (3.6%)	4 (7.3%)	4 (7.3%)	0 (0.0%)	0 (0.0%)	0 (0.0%)	0 (0.0%)	55 (8.3%)
2013–14	10 (16.4%)	24 (39.3%)	2 (3.3%)	0 (0.0%)	7 (11.5%)	15 (24.6%)	0 (0.0%)	1 (1.6%)	2 (3.3%)	61 (9.2%)
2014–15	5 (6.8%)	22 (30.1%)	5 (6.8%)	0 (0.0%)	26 (35.6%)	11 (15.1%)	1 (1.4%)	0 (0.0%)	3 (4.1%)	73 (11.0%)
2015–16	8 (21.1%)	10 (26.3%)	2 (5.3%)	0 (0.0%)	0 (0.0%)	17 (44.7%)	0 (0.0%)	1 (2.6%)	0 (0.0%)	38 (5.7%)
2016–17	28 (49.1%)	0 (0.0%)	1 (1.8%)	3 (5.3%)	1 (1.8%)	14 (24.6%)	2 (3.5%)	2 (3.5%)	6 (10.5%)	57 (8.6%)
2017–18	8 (25.0%)	15 (46.9%)	0 (0.0%)	2 (6.3%)	2 (6.3%)	1 (3.1%)	0 (0.0%)	2 (6.3%)	2 (6.3%)	32 (4.8%)
Tot *	92 (13.9%)	268 (40.4%)	35 (5.3%)	14 (2.1%)	89 (13.4%)	92 (13.9%)	13 (2.0%)	41 (6.2%)	19 (2.9%)	663 (100%)

Column (*) and row (**) totals. Raw percentages are reported, except for the last column with number of samples for each season.

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
