# Peer review of "10-Year Rotavirus Infection Surveillance: Epidemiological Trends in the Pediatric Population of Perugia Province"

_ijerph, 2020, doi:10.3390/ijerph17031008_

Round 1

Reviewer 1 Report

The manuscript makes a contribution to worldwide rotavirus surveillance. It covers RVA epidemiology in central Italy in pediatric population during 10 years. The genotype distribution is described with the emergence of G12P[8] strains during the reported period.

The manuscript is written comprehensibly and the results are well documented by two figures and two tables. There are just some minor spelling mistakes or typos which should be corrected.

P5, L164: “RAVGE” correctly it is RVAGE.

P5, L177: “...second year in in Europe...” – omit the extra preposition.

P5, L184: “Collected data represents...” – correctly it should be “represent”.

Next, I have a question about the mixed infections reported both in the Fig. 2 and Table 2. The numbers of samples with multiple genotypes was quite high (total n=41) thus I think the list of mixed genotypes should be included in the Results chapter. Also the uncommon genotypes should be listed namely as those are definitely of great interest.

Author Response

The manuscript makes a contribution to worldwide rotavirus surveillance. It covers RVA epidemiology in central Italy in pediatric population during 10 years. The genotype distribution is described with the emergence of G12P[8] strains during the reported period.

The manuscript is written comprehensibly and the results are well documented by two figures and two tables. There are just some minor spelling mistakes or typos which should be corrected.

P5, L164: “RAVGE” correctly it is RVAGE.

P5, L177: “...second year in in Europe...” – omit the extra preposition.

P5, L184: “Collected data represents...” – correctly it should be “represent”.

Next, I have a question about the mixed infections reported both in the Fig. 2 and Table 2. The numbers of samples with multiple genotypes was quite high (total n=41) thus I think the list of mixed genotypes should be included in the Results chapter. Also the uncommon genotypes should be listed namely as those are definitely of great interest.

Answer: we thank you the Reviewer for identified spelling mistakes and typos. We have corrected them. As for the question, we have included the list of uncommon genotypes found (“including G10P[8] (n=4, 0.6%), G2P[8] (n=3, 0.5%), G4P[4] (n=2, 0.3%), G1P[4] (n=1, 0.2%), G4P[10] (n=1, 0.2%), G6P[9] (n=1, 0.2%) and G9P[4] (n=1, 0.2%).”). Furthermore, a sentence has been included in order to report the distribution of genotypes in mixed infections (“As for mixed infections, 12 different mixed genotypes were identified in 41 samples. Among them the most common were G1G9P[8] (n=12, 1.8%), G1G2P[4]P[8] (n=11, 1.7%), G1G4P[8] (n=6, 0.9%), G9G12P[8] (n=3, 0.5%) and G2G9P[4]P[8] (n=2, 0.3%). G1G12P[8], G1G2P[4],  G1G3P[4]P[8], G2G12P[4]P[8], G2G4P[4], G3G9P[8], G4G9P[8] were identified in one sample each (0.2%).”)

Reviewer 2 Report

1. General remark

Three kinds of protocols have been proposed by ECDC for follow up of rotavirus gastroenteritis in relationship with vaccination introduction. The protocol “ECDC TECHNICAL DOCUMENT Impact of rotavirus vaccination Generic study protocol” is the closest to the approach of the investigators. When looking to the proposed methodology, the one followed by the investigators is at best introductory. Essential elements are missing, such as “number of acute gastroenteritis” in the province. The authors should look carefully to the proposed protocol and complete their very preliminary analysis when possible. 

2. Abstract lines 17-18

“outpatients < 15 years of age presenting with gastroenteritis to the hospitals of the Perugia province. »
Term « outpatients » Not clear: includes « outpatients » and « hospitalised patients”: which proportion was exclusively outpatient and which complementary proportion was attending as outpatient, and then hospitalized?
3. Material & Methods line 99

“if then admitted to hospital for short observation or for requiring further treatment as inpatients.”
“Admitted to hospital for short observation”: is it “one-day care”? Were simple medical consultations for gastro-enteritis excluded from the numerator? Were emergency medical department consultations (not followed by inpatient hospitalisation) excluded?
4. Results line 130

“Hospitalisation rate per 100 000 inhabitants in the paediatric population of the province of Perugia.”

Be more precise: suggestion: Hospitalisation rate (including one-day care) for rotavirus associated gastroenteritis per 100 000 inhabitants in the 0 to 15-year-old population.

5. Lines 146-147:

“Figure 2. Absolute frequencies of the genotypes detected (see legend) and hospitalization rate per 100,000 inhabitants stratified by surveillance season. »

Be more precise : suggestion : … of the rotavirus genotype…

6. Lines 185-187

“This is even more true because the paper addressed community acquired cases that are expected to be mostly impacted.”

No: you cannot infer community data on rotavirus associated gastroenteritis from your data, as you included only hospitalised inpatients and subjects admitted to hospital for short observation.

7. Figure 1

Redundant with data in Table 1: please choose on form of presentation Table OR Figure, without redundancy.

Author Response

Comments and Suggestions for Authors

General remark

Three kinds of protocols have been proposed by ECDC for follow up of rotavirus gastroenteritis in relationship with vaccination introduction. The protocol “ECDC TECHNICAL DOCUMENT Impact of rotavirus vaccination Generic study protocol” is the closest to the approach of the investigators. When looking to the proposed methodology, the one followed by the investigators is at best introductory. Essential elements are missing, such as “number of acute gastroenteritis” in the province. The authors should look carefully to the proposed protocol and complete their very preliminary analysis when possible. 

Answer: we thank the Reviewer for this important suggestion. We did not perform a pre-post study as correctly suggested by the ECDC protocol as the vaccination in Umbria Region has started just recently and coverage is quite low. Indeed, we did not follow the protocol even though the rate of rotavirus cases is one of the indicators suggested for the evaluation of the impact of vaccination. We acknowledge that further indicators should be taken into consideration in order to have a broader view and we will take it into account for future research. In the meantime, we have included in the limits/strengths section a reference to your point.

Abstract lines 17-18

“outpatients < 15 years of age presenting with gastroenteritis to the hospitals of the Perugia province. » Term « outpatients » Not clear: includes « outpatients » and « hospitalised patients”: which proportion was exclusively outpatient and which complementary proportion was attending as outpatient, and then hospitalized?
Answer: we thank the Reviewer for the requested clarification. We have modified the abstract in order to make it clearer the study population (“Between September 2007 and August 2018, 663 RV-positive stool specimens were collected from children < 15 years of age presenting with gastroenteritis to the emergency room of Perugia province hospitals who were then hospitalized.”). In order to stick to the word limits further minor amendments have been subsequently made.

Material & Methods line 99

“if then admitted to hospital for short observation or for requiring further treatment as inpatients.”
“Admitted to hospital for short observation”: is it “one-day care”? Were simple medical consultations for gastro-enteritis excluded from the numerator? Were emergency medical department consultations (not followed by inpatient hospitalisation) excluded?

Answer: thank you for your request. We have made the sentence clearer as follows “Children below 15 years of age […] were considered eligible for inclusion in the study if then admitted to the hospital as inpatients, including one-day care (at least one overnight stay). Nosocomial infections, namely those occurring in inpatients after at least 48 hours from hospital admission, were excluded as well as cases requiring emergency medical department consultations not followed by inpatient hospitalization.

Results line 130

“Hospitalisation rate per 100 000 inhabitants in the paediatric population of the province of Perugia.”

Be more precise: suggestion: Hospitalisation rate (including one-day care) for rotavirus associated gastroenteritis per 100 000 inhabitants in the 0 to 15-year-old population.

Answer: the caption has been modified according to the reviewer’s suggestion.

Lines 146-147:

“Figure 2. Absolute frequencies of the genotypes detected (see legend) and hospitalization rate per 100,000 inhabitants stratified by surveillance season. »

Be more precise : suggestion : … of the rotavirus genotype…

Answer: the caption has been modified according to the reviewer’s suggestion.

Lines 185-187

“This is even more true because the paper addressed community acquired cases that are expected to be mostly impacted.”

No: you cannot infer community data on rotavirus associated gastroenteritis from your data, as you included only hospitalised inpatients and subjects admitted to hospital for short observation.

Answer: we thank the Reviewer for this observation. We are aware of it and we have already reported in the limits section that we cannot address the overall burden of RVAGE with our work. Nonetheless, we have modified the sentence as follows “In fact, even though this paper does not release a thorough overview of all community acquired cases, it is expected that vaccination will also impact on hospitalizations due to RVAGE.”

Figure 1

Redundant with data in Table 1: please choose on form of presentation Table OR Figure, without redundancy.

Answer: following the Reviewers’ suggestion, we have removed figure 1.

Reviewer 3 Report

The manuscript that I reviewed “10-Year Rotavirus Infection Surveillance: Epidemiological Trends in the Pediatric Population of Perugia Province” is a study aimed to perform an epidemiological and molecular surveillance on Rotavirus A infections by collecting pre-vaccination data in the pediatric population of the Perugia province in Umbria Region, Italy. A total of 663 RVA-positive stool specimens were collected, during 11 RVA surveillance seasons, from outpatients < 15 years of age presenting with gastroenteritis to the hospitals of the Perugia, between September 2007 and August 2018. The hospitalization rate resulted higher in children aged 5 years or less compared to the whole sample and higher in males compared to females. Regarding the genotyping for VP7 and VP4, the G1P resulted the most prevalent among the common genotypes and also in 7 of 11 seasons examined. Other genotypes found with greater proportions were G9P, G4P and G12P.

Comments to the Authors:

Overall, the study conducted is of interest since adds pre-vaccination epidemiological data of the pediatric population of a province in Umbria Region, Italy. The manuscript is well written as a whole, detailed and supported by presented data. I have only some observations.

Line 44: at the end of the line there is twice the reference number “8”.

Line 53: there is a dash before the parenthesis end.

Line 70: there is a comma at the end of the sentence.

Line 87: I suggest to the Authors to indicate the name of the regional reference center.

Line 90: there are two commas.

Line 113: I suggest to the Authors to add the number of samples collected and not only the percentage. For example, 59,1% (392/663).

Line 116-120: I suggest adding also the reference of Table 1 and not only Figure 1, since the percentages described are also reported in the table 1.

Table 1: the number “89.7%” in the season 2010-2011 is highlighted.

Line 139: I suggest adding also at the end of this sentence “(Figure 2; Table 2)”.

Table 2: I suggest to the Authors to add a final column with the total of the genotypes identified in each season in order to clarify how the Authors obtained the percentages indicated in the table.

Author Response

The manuscript that I reviewed “10-Year Rotavirus Infection Surveillance: Epidemiological Trends in the Pediatric Population of Perugia Province” is a study aimed to perform an epidemiological and molecular surveillance on Rotavirus A infections by collecting pre-vaccination data in the pediatric population of the Perugia province in Umbria Region, Italy. A total of 663 RVA-positive stool specimens were collected, during 11 RVA surveillance seasons, from outpatients < 15 years of age presenting with gastroenteritis to the hospitals of the Perugia, between September 2007 and August 2018. The hospitalization rate resulted higher in children aged 5 years or less compared to the whole sample and higher in males compared to females. Regarding the genotyping for VP7 and VP4, the G1P resulted the most prevalent among the common genotypes and also in 7 of 11 seasons examined. Other genotypes found with greater proportions were G9P, G4P and G12P.

Comments to the Authors:

Overall, the study conducted is of interest since adds pre-vaccination epidemiological data of the pediatric population of a province in Umbria Region, Italy. The manuscript is well written as a whole, detailed and supported by presented data. I have only some observations.

Line 44: at the end of the line there is twice the reference number “8”.

Answer: the reference “8” is not reported twice. The first “8” refers to P-genotype.

Line 53: there is a dash before the parenthesis end.

Answer: thank you for the warning. We have removed the dash before the parenthesis.

Line 70: there is a comma at the end of the sentence.

Answer: thank you for the warning. We have eliminated the comma at the end of the sentence.

Line 87: I suggest to the Authors to indicate the name of the regional reference center.

Answer: the regional center belongs to the University of Perugia. The information has been included in the paper.

Line 90: there are two commas.

Answer: thank you for the warning. We have eliminated one of the two commas.

Line 113: I suggest to the Authors to add the number of samples collected and not only the percentage. For example, 59,1% (392/663).

Answer: thank you for the suggestion. We have added the absolute number of samples collected (“59.1% (392/663) of the sample were collected from male children […]; 88.8% (589/663) of the samples were collected from children below 5 years of age”). We have also included the information on absolute numbers in the remaining part of results.

Line 116-120: I suggest adding also the reference of Table 1 and not only Figure 1, since the percentages described are also reported in the table 1.

Answer: the reference to Table 1 has been included also because, following another Reviewer’s suggestion, figure 1 has been removed as redundant.

Table 1: the number “89.7%” in the season 2010-2011 is highlighted.

Answer: thanks for the warning. We have removed the highlighting.

Line 139: I suggest adding also at the end of this sentence “(Figure 2; Table 2)”.

Answer: thank you for the suggestion. The reference to Table 2 and current Figure 1 has been included. 

Table 2: I suggest to the Authors to add a final column with the total of the genotypes identified in each season in order to clarify how the Authors obtained the percentages indicated in the table.

Answer: a final column reporting the total for each season has been included. Furthermore, it has been specified that raw percentages have been reported except for the last new column with number of samples for each season.

Reviewer 4 Report

De Waure et al have evaluated the prevalence and epidemiological trends of rotavirus in Perugia Province, Italy. It is an important study since rotavirus is the major cause of acute gastroenteritis among children.

However, the manuscript needs a few improvements before publishing.

Introduction and discussion may be restructured to improve the readability.

Line 29-37: All the sentences are about the same point. It can be summarized. Line 30-31 is included in line 32-34 sentences.

Line 40, 42: RV is defined as rotavirus. So, "RV are" is not suitable.

Line 47: Remove ‘

Line 70: Put “.”

Line 78: September 1st and August 31st

Line 88: Add space after (ISS)

Line 90: Remove the “,”

Figure 1: There is a decreasing trend in hospitalizations. It is better to explain the reason in the discussion.

Line 132-139: What are the possible reasons for this observation.

Line 173-174: Any statistical evidence?

Line 183: Add a “.”

Author Response

De Waure et al have evaluated the prevalence and epidemiological trends of rotavirus in Perugia Province, Italy. It is an important study since rotavirus is the major cause of acute gastroenteritis among children.

However, the manuscript needs a few improvements before publishing.

Introduction and discussion may be restructured to improve the readability.

Answer: thank you for the suggestion. We have rephrased several parts and included some amendments in order to make the text to flow.

Line 29-37: All the sentences are about the same point. It can be summarized. Line 30-31 is included in line 32-34 sentences.

Answer: the sentences have been merged together. Nevertheless, worldwide and European data have been left in order to give a thorough overview of the problem.

Line 40, 42: RV is defined as rotavirus. So, "RV are" is not suitable.

Answer: thank you for the warning. We have replaced RV with RVs.

Line 47: Remove ‘

Answer: thank you for the warning. We have corrected the typo.

Line 70: Put “.”

Answer: thank you for the warning. We have corrected the typo.

Line 78: September 1st and August 31st

Answer: thank you for the warning. We have corrected the typo.

Line 88: Add space after (ISS)

Answer: thank you for the warning. We have corrected the typo.

Line 90: Remove the “,”

Answer: thank you for the warning. We have corrected the typo.

Figure 1: There is a decreasing trend in hospitalizations. It is better to explain the reason in the discussion.

Answer: thank you for this useful suggestion. We have included a new paragraph reporting one of most likely reason (in our opinion) for the observed decreasing trend (“Albeit we did not apply any method to assess the significance of temporal trends, a decreasing trend in hospitalization rates can be appreciated. This could be due to the increasing attention into the appropriateness of the management of AGE. In fact, the hospitalization rate for AGE in childhood has been used and is currently used as an indicator to assess the quality of primary care in Italy.”)

Line 132-139: What are the possible reasons for this observation.

Answer: the percentage of non-typable samples and mixed infections was aligned with other data. A sentence on non-typable samples was already in the discussion, but we have amended it in order to include also mixed infections. For this reason, we have also included other references at the end of the sentence (“As for the genotype, the percentage of non-typable samples and mixed infections followed available data [10, 22, 38]”). For the sake of clarity, nontypeable samples may be due to the low virus concentration in stool samples, but it cannot be also excluded that some samples contain either unusual G or P types or mutations in the sequences targeted by PCR primers. On the other hand, mixed infections depend on the frequency of each strain circulating in the population and possible interference phenomena during replication in the gut. Nevertheless, these explanations have not included in the text as not relevant – in our opinion - for the purpose of the study. As for the G1P[8] and G9P[8]  discussion already elaborated on that as they were the most common identified genotypes also at European level. Eventually, the variability of genotypes distribution is intrinsic to RV infections supporting also virological surveillance.

Line 173-174: Any statistical evidence?

Answer: we did not investigate the statistical significance of the difference in rates. Indeed, we have just reported that rates were higher in males. Nevertheless, references reported at the end of the sentence have addressed the significance of differences between sexes.

Line 183: Add a “.”

Answer: thank you for the warning. We have corrected the typo.

Round 2

Reviewer 2 Report

More precision has been brought on the type of patients included in your prospective follow-up study. It now clearly and prercisely describes the children attending ED for gastro-enteritis and those being hospitalised. The criteria of inclusion varies between various studies, and comparisons are difficult. This limitation is also clearly descibed in their discussion. Difficult to harmonize such studies at the European level (illusion of harmonization), and it is useful to get some precise descriptions of the "real world" inclusion criteria in some studies as this one.